

# Analysis of the differences in physicochemical properties, volatile compounds, and microbial community structure of pit mud in different time spaces

Baolin Han[1], Hucheng Gong[1], Xiaohu Ren[1], Shulin Tian[1], Yu Wang[1], Shufan Zhang[1], Jiaxu Zhang[2] and Jing Luo[2]

[1] College of Bioengineering, Sichuan University of Science & Engineering, Yibin, Sichuan, China
[2] Chengdu Shuzhiyuan Liquor Industry Co., Ltd, Chengdu, Sichuan, China

Corresponding author
Baolin Han, han_bl1986@163.com

## ABSTRACT

Pit mud (PM) is among the key factors determining the quality of Nongxiangxing baijiu, a Chinese liquor. Microorganisms present inside PM are crucial for the unique taste and flavor of this liquor. In this study, headspace solid-phase microextraction was used in combination with gas chromatography and high-throughput sequencing to determine the volatile compounds and microbial community structure of 10- and 40-year PM samples from different spaces. The basic physicochemical properties of the PM were also determined. LEfSe and RDA were used to systematically study the PM in different time spaces. The physicochemical properties and ester content of the 40-year PM were higher than those of the 10-year PM, but the spatial distribution of the two years PM samples exhibited no consistency, except in terms of pH, available phosphorus content, and ester content. In all samples, 29 phyla, 276 families, and 540 genera of bacteria, including four dominant phyla and 20 dominant genera, as well as eight phyla, 24 families, and 34 genera of archaea, including four dominant phyla and seven dominant genera, were identified. The LEfSe analysis yielded 18 differential bacteria and five differential archaea. According to the RDA, the physicochemical properties and ethyl caproate, ethyl octanoate, hexanoic acid, and octanoic acid positively correlated with the differential microorganisms of the 40-year PM, whereas negatively correlated with the differential microorganisms of the 10-year PM. Thus, we inferred that *Caproiciproducens*, *norank_f__Caloramatoraceae*, and *Methanobrevibacter* play a dominant and indispensable role in the PM. This study systematically unveils the differences that affect the quality of PM in different time spaces and offers a theoretical basis for improving the declining PM, promoting PM aging, maintaining cellars, and cultivating an artificial PM at a later stage.

## INTRODUCTION

Chinese baijiu is among the six major distilled liquors in the world. It is not only popular in China but also enjoys a certain status abroad (*Tu et al., 2022*; *Wei et al., 2020*). This liquor is typically made through solid-state fermentation of different grains in open and multi-micro coexistence environmental conditions. Based on its different production processes and flavor characteristics, baijiu can be divided into four basic aroma types: Nongxiangxing, Jiangxiangxing, Qingxiangxing, and Mixiangxing, as well as into eight aroma types derived from the basic aroma types (Jianxiangxing, Fuyuxiangxing, Texiangxing, Fengxiangxing, Dongxiangxing, Chixiangxing, Zhimaxiangxing, and Laobaiganxiangxing) (*Chen et al., 2022*; *Tong et al., 2021*). Among these types, the annual output of Nongxiangxing baijiu is >70% of China's baijiu market. This type of baijiu is popular among people for its unique taste and flavor (*Yuan et al., 2022*). Fermentation carried out in the mud cellar is the most typical feature of Nongxiangxing baijiu production. This is the key to differentiating this type of baijiu from the other types of spiced baijiu. The quality of the put mud (PM) is among the key factors determining the quality of Nongxiangxing baijiu (*Chai et al., 2019*). The PM is composed of special soil, bran koji powder, yellow water, hexanoic acid bacterial solution, liquor tail, *etc.* It contains bacteria, archaea, and fungi, and is the seat of microbial growth and reproduction during Nongxiangxing baijiu production. If PM quality is inferior, some key brewing and aroma-producing microorganisms grow, reproduce, and metabolize obstruct, thereby lacking the necessary key volatile compounds (*Gao, Wu & Zhang, 2020*).

PM quality is significantly different between old and new cellars because of different years of use, and the content of volatile compounds, such as ethyl caproate and ethyl acetate, in the produced Nongxiangxing baijiu exhibits certain differences. According to numerous studies, baijiu produced in old cellars has a higher content of ethyl caproate and a lower content of ethyl acetate, whereas its flavor is more prominent, the body is better, and quality is superior than those produced in new cellars (*Xu et al., 2022a*). Because old cellars have been continuously used for years, their internal microbes are constantly domesticated and form some microbiome, which is a crucial player in the key node position in Nongxiangxing baijiu production and is the vital factor affecting PM quality (*Zhou et al., 2021*). Moreover, the physicochemical properties, volatile compounds, and microbial community structure of PM in different cellar ages and locations also differ to some extent. The complex micro-ecological environment inside the cellar and artificial cellar maintenance are the primary reasons for these differences (*Yan et al., 2023*). According to the evaluation system employed, these three indicators can be used as standards for evaluating PM quality in certain weights. The correlation among these three indicators is the most powerful embodiment for distinguishing PM quality because these indicators affect each other and jointly maintain PM stability. Moisture, available phosphorus, and available potassium were key factors affecting the contents of *Clostridium* spp., hexanoic acid, and ethyl caproate in PM (*Wu et al., 2022a*). *Liu et al. (2018)* screened the physicochemical properties related to the microbial community structure based on PM quality. They found that total nitrogen, ammoniacal nitrogen, and pH can be used

as the preliminary criteria for evaluating PM quality. Although numerous studies have reported significant differences between old and new PM and some evaluation criteria for PM quality are preliminatively determined through correlation analysis, a clear definition of the key microorganisms that cause the differences and affect PM quality is lacking.

In the anaerobic fermentation system, some key microorganisms with obvious differences in relative abundances are present in the new and old PM, and these microbes can significantly affect PM quality. Such as *Caproiciproducens* and *Syntrophomonas* have a significant correlation is observed with the main volatile compounds. Therefore, finding the key microbes is the top priority in solving the problems related to Nongxiangxing baijiu development. *Gao et al. (2022)* recently constructed a simplified microbiome that was associated with key volatile compounds, thereby increasing the ester content of upper fermented grains. The relative abundance of *Syntrophomonas* is higher in the old PM, and *Clostridium* can produce butyric acid and hexanoic acid. *Sun et al. (2022)* found a mutualistic mechanism between *Clostridium* and *Syntrophomonas* by co-culturing their two key strains isolated from the PM. This mechanism could increase the accumulation of butyric and hexanoic acids. On exploring the evolutionary patterns and driving factors of the microbial community structure of PM, *Chai et al. (2021)* found that *Caproiciproducens* and *Methanosarcina* were the key microbial genera in the PM as the cellar age increased and were significantly and positively correlated with major volatile compounds. Studies have demonstrated that the abundance of *Lactobacillus* is high in a new PM, which is the signature genus (*Ren et al., 2018*). Moderate abundance of *Lactobacillus* can promote the domestication of beneficial microbes in the PM, whereas excessive abundance destroy the micro-ecological balance of the PM (*Zhao et al., 2020*). Thus, key microbes play a vital role in the PM.

Based on these results, this study considered the PM of an enterprise in Qionglai as the research object. By determining the physicochemical properties, volatile compounds, and microbial community structure of the PM, and comprehensively comparing and analyzing the connection and difference between PM from different time spaces, we determined the key indicators and microorganisms in the PM. We aimed to provide some theoretical reference for improving the declining PM, maintaining cellars, and cultivating an artificial PM in the later stage, so as to enhance the quality of PM and Nongxiangxing baijiu.

# MATERIALS & METHODS
## Sampling
The PM samples were provided by Shu Zhiyuan Enterprise in Qionglai City of Sichuan Province. Three cellars were selected for parallel sampling at 10- and 40-year old each PM, and the upper PM (50 cm from the cellar entrance), middle PM (1/2 of the cellar wall), and lower PM (50 cm from the cellar bottom) were immediately mixed from each cellar wall, respectively. Fig. S1 presents the sampling position of each cellar wall. The PM samples were correspondingly named U10 (U10_1–U10_3), M10 (M10_1–M10_3), L10 (L10_1–L10_3), U40 (U40_1–U40_3), M40 (M40_1–M40_3), and L40 (L40_1–L40_3). A total of 18 samples were collected. Then, 200 g of each mixed sample was collected, frozen

at a low temperature in a sterile bag, and transported back to the laboratory for subpacking. One sample was stored at −20 °C and used for determining physicochemical properties and volatile compounds. Another sample was stored at −80 °C and used for microbial sequencing.

## Physicochemical properties

Moisture, ammoniacal nitrogen, humus, and available phosphorus were determined using the drying method, Nassler reagent (Tianjin Zhiyuan Chemical Reagent Co., Ltd., Tianjin, China) colorimetric method, potassium dichromate (Tianjin Zhiyuan Chemical Reagent Co., Ltd., Tianjin, China) oxidation method, and ammonium molybdate (Tianjin Zhiyuan Chemical Reagent Co., Ltd., Tianjin, China) colorimetric method, respectively. All these factors were determined by referring to *Shen*'s (*2014*) "Complete Book of Baijiu Production Technology". pH was determined using the pH meter method by referring to the method by *Zhang et al. (2020a)*.

## Volatile compounds

Volatile compounds were determined through headspace solid-phase microextraction coupled with gas chromatography-mass spectrometry (HS-SPME-GC-MS) with slight modifications from *Bi et al. (2022)*. First, 1 g of PM was accurately weighed and added to a headspace vial. Then, 2 g of NaCl, five mL of ultrapure water, and 10 µL of 41.1 µg/g 2-octanol (Chengdu Colon Chemical Co., Ltd., Chengdu, China) internal standard were added to the PM, sonicated for 10 min, and equilibrated at 50 °C for 15 min. Next, 50/30 µm DVB/CAR/PDMS fiber extraction head was inserted at the upper 1–2 cm of the liquid surface for 30 min, removed and inserted into the injection port, and resolved at 230 °C for 3 min to perform the GC-MS analysis. The sample mass spectrometry data were compared with data in the standard universal library (NIST2005), and substances with a match of >800 were selected for semi-quantitative analysis using the internal standard method.

Chromatography-mass spectrometry was performed using the J&W 122-7062 column (60 m × 250 µm × 0.25 µm). Ramp-up procedure: the sample was holded at 40 °C for 5 min. The column temperature was then increased to 100 °C at 4 °C/min, followed by a further increase to 230 °C at 6 °C /min. This temperature was maintained for 10 min, and the inlet temperature was 230 °C. High-purity helium was used as the carrier gas, without split injection, and the flow rate was 1.0 mL/min. The temperatures of the ion source and connecting line were 230 °C and 250 °C, respectively, the EI electron energy was 70 eV, and the mass spectrometry scan range was 35–400 amu.

## Microbial community composition

Genomic DNA was extracted from the PM samples by using the D5625-00 Soil DNA Extraction Kit (Omega Biotek, Norcross, GA, USA). Extraction was performed by referring to the method described in the kit, and the DNA extract quality was measured through 2%(2 g/100 mL) agarose gel electrophoresis (Beijing Liuyi Bio Co., Ltd., Beijing, China). Bacterial primers 338F (5′-ACTCCTACGGGAGGCAGCAG-3′) and 806R (5′-GGACTACHVGGGTWTCTAAT-3′),

and Arch344F (5′-ACGGGGYGCAGCAGGCGCGA-3′) and Arch915R (5′-GTGCTCCCCCGCCAATTCCT-3′) were used for PCR amplification of the V3-V4 and ARC variable regions, respectively. The amplification procedure was as follows: pre-denaturation at 95 °C for 3 min; 35 cycles of denaturation at 95 °C for 30 s, annealing at 55 °C for 30 s, and extension at 72 °C for 45 s; stable extension at 72 °C for 10 min, and finally storage at 10 °C until the reaction ceased (*Zhang et al., 2014*). Shanghai Meiji Bio Ltd., China was commissioned to perform sequencing by using the Illumina MiSeq PE250 platform, and the raw data were uploaded to the NCBI database.

## Data analysis

All raw 16S rRNA gene sequencing data were generated through high-throughput sequencing and processed using QIIME (V1.9.1). After the data were subjected to initial quality control (QC) processing, the reads were merged using FLASH (V1.2.11) to obtain the raw tags, which were then filtered using the QIIME function to obtain clean tags (*Caporaso et al., 2010*). The chimeras were detected and removed using the Uparse algorithm to obtain valid tags, which were grouped into the same operational taxonomic units (OTUs) at a 97% similarity threshold. Representative sequences of each OTU were extracted and annotated using the SILVA database V138 (*Quast et al., 2013*).

Community richness and diversity were determined using ACE and Chao1 indices and Sobs and Shannon indices, respectively. The community composition of the PM samples was compared using bar maps. SIMCA14.1 was used for performing PLS-DA of volatile compounds. The Linear discriminant analysis Effect Size (LEfSe) was applied to evaluate differential microbes, and Canoco 5 was used to analyze the correlations between physicochemical properties, differential volatile compounds, and microbial communities. For all statistical analyses, significance was determined using IBM SPSS Statistics software (V27.0). $P < 0.05$ was considered statistically significant.

# RESULTS

## Physicochemical properties of the PM

Table 1 presents the physicochemical properties of different PM samples. For the same-level samples at different cellar ages, the physicochemical properties of the 40-year PM were higher than those of the 10-year PM. For the samples of the same cellar age but at different levels, the moisture and pH of the 10-year PM decreased first from top to bottom and then increased, and the ammoniacal nitrogen, humus, and available phosphorus contents gradually increased. The moisture and humus of the 40-year PM increased first from top to bottom and then decreased. The pH and ammoniacal nitrogen content first decreased and then increased, and no significant difference was observed in the ammoniacal nitrogen content. By contrast, the content of available phosphorus increased gradually.

## Volatile compounds of the PM

As shown in Fig. 1A, 40 volatile compounds were detected in all PM samples, namely 19 esters, 11 acids, four alcohols, one ketones, three phenols, and two others.

As shown in Fig. 1B and Table S2, the content of esters was higher in the 40-year PM than in the 10-year PM, and the content increased in both PM samples as the cellar
**Table 1   Physicochemical properties of different PM samples.**

| PM samples | Moisture (%) | pH | Ammoniacal nitrogen (mg/100 g) | Humus (%) | Available phosphorus (mg/100 g) |
|---|---|---|---|---|---|
| U10 | $36.52 \pm 0.72^d$ | $4.72 \pm 0.10^c$ | $9.74 \pm 0.22^c$ | $9.63 \pm 0.08^f$ | $106.10 \pm 4.18^e$ |
| M10 | $36.04 \pm 0.42^d$ | $3.62 \pm 0.01^e$ | $11.41 \pm 0.60^{bc}$ | $11.18 \pm 0.13^e$ | $130.87 \pm 8.00^{de}$ |
| L10 | $38.50 \pm 0.11^{cd}$ | $3.67 \pm 0.01^e$ | $14.53 \pm 0.71^b$ | $12.95 \pm 0.09^d$ | $151.59 \pm 4.75^d$ |
| U40 | $41.07 \pm 3.45^{bc}$ | $6.72 \pm 0.45^a$ | $42.15 \pm 0.66^a$ | $16.43 \pm 0.10^b$ | $403.20 \pm 18.65^c$ |
| M40 | $51.30 \pm 0.81^a$ | $4.33 \pm 0.10^d$ | $38.69 \pm 4.11^a$ | $21.43 \pm 0.07^a$ | $486.18 \pm 32.86^b$ |
| L40 | $43.57 \pm 0.56^b$ | $5.50 \pm 0.09^b$ | $38.78 \pm 1.03^a$ | $16.11 \pm 0.10^c$ | $571.07 \pm 28.07^a$ |

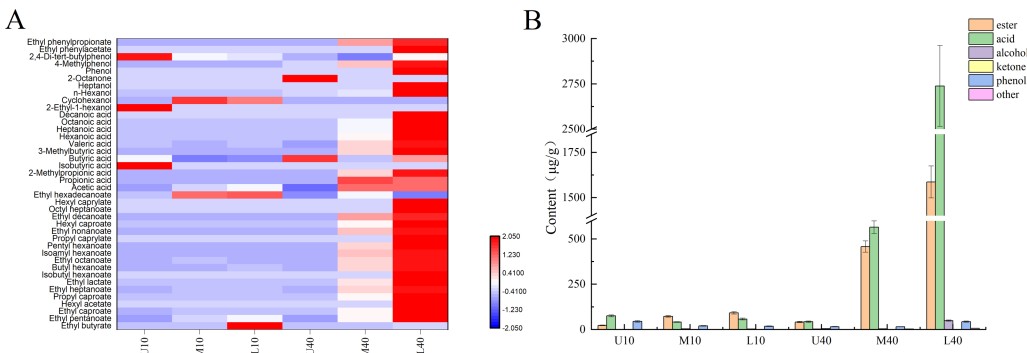

**Figure 1   Heat map of volatile compounds in PM (A), volatile compounds content (B).**

depth increased. M40 and L40 samples had significantly higher ester content than the other samples. The content of major esters such as ethyl caproate, ethyl heptanoate, butyl caproate, ethyl octanoate, and hexyl caproate was higher in the 40-year PM than in the 10-year PM and increased with an increase in the cellar depth, whereas the ethyl hexadecanoate content was higher in the 10-year PM than in the 40-year PM. This indicated that the ester content increased as well as decreased with an increase in the cellar age, which was determined by the micro-ecological environment inside the cellar. The highest ethyl caproate content in L40 was 1,053.54 µg/g, and the ethyl caproate content in L10 and L40 was 4.45 and 37.19 times higher than that in U10 and U40, respectively. The ethyl caproate content in the 40-year PM was 13.63 times higher than that in the 10-year PM. The acid content was higher in M40 and L40 than in M10 and L10. The spatial distribution of the acid content in the 10-year PM exhibited a decreasing trend, followed by an increasing trend, whereas the spatial distribution of the 40-year PM increased with an increase in the cellar depth. At the same time, the acid content in M40 and L40 was significantly higher than that of the other samples. The contents of acetic acid, butyric acid, hexanoic acid, heptanoic acid, octanoic acid, and other major acid volatile compounds were higher in M40 and L40 than in M10 and L10. The spatial distribution of hexanoic acid exhibited a gradually increasing trend in the 40-year PM. The highest content of hexanoic and octanoic acids in L40 was 1,669.48 and 727.15 µg/g, respectively. The contents of hexanoic and octanoic acids in L40 were 350.73

and 341.38 times higher than those in U40, respectively. In the meantime, the contents of hexanoic and octanoic acids in the 40-year PM were 24.09 and 28.80 times higher than those in the 10-year PM. The alcohol content was higher in the 40-year PM than in the 10-year PM, and the spatial distribution of the alcohol content in the 10-year PM exhibited an increasing trend, followed by a decreasing trend, whereas the spatial distribution of the alcohol content in the 40-year PM exhibited a gradually increasing trend. The alcohol content was significantly higher in L40 than in the other samples. The main flavors of alcohol were cyclohexanol and n-hexanol. The content of n-hexanol in the 40-year PM increased with an increase in the cellar depth. The ketones included 2-octanone; phenols included phenol, 4-methylphenol, and 2,4-di-tert-butylphenol; and other flavors included ethyl phenylacetate and ethyl phenylpropionate aromatic hydrocarbon compounds.

## Microbial community structure of the PM

In total, 18 samples were sequenced using the high-throughput sequencing platform for bacterial amplicons, and 1,324,264 valid bacterial sequences with an average length of 411 bp were obtained. The optimized sequences were grouped according to the default value of 97%, and 2,302 OTUs were obtained. The 18 samples were then sequenced with archaeal amplicons, and 1,584,203 valid archaeal sequences with an average length of 224 bp were obtained. The optimized sequences were grouped according to the default value of 97%, and 494 OTUs were obtained. The coverage rate of each sample was >99%, which indicated that the number of sequenced sequences was sufficient and could represent the microbial community composition. The curves of different PM samples tended to parallel straight lines and close to saturation once the sequencing depth reached 25,000 (Figs. S2A and S2B). This indicated that the experimental results were true and reliable. The PM samples used basically represented the community composition of all microorganisms in the samples.

## Alpha diversity of the PM

As shown in Table S2A, the $\alpha$-diversity index of bacteria was higher in the 10-year PM than in the 40-year PM, whereas the spatial distribution basically exhibited a decreasing trend in the 10-year PM and an increasing trend in the 40-year PM.

As shown in Table S2B, the overall trend of the $\alpha$-diversity index of archaea in terms of cellar age and spatial location was consistent with that of bacteria, decreased with increasing cellar age. However, the $\alpha$-diversity index decreased with increasing cellar depth in the 10-year PM, whereas increased with increasing cellar depth in the 40-year PM.

The levels of bacterial genera in different PM samples exhibited some differences (Fig. 2A). A total of 81 common microbial genera were observed in all samples, and the specific microbial genera were higher in the 10-year PM than in the 40-year PM, with most genera (79) being specific to U10. The other samples M10–L40 had 19, 52, four, 10, and three specific genera, respectively. The results were consistent with those of the $\alpha$-diversity index analysis, which unveiled that the 10-year PM had numerous microorganisms because of its short service life. Thus, the 10-year PM was not domesticated and transformed into the dominant microbiome, resulting in more specific genera. By contrast, the 40-year PM was domesticated and subjected to stress for a long time, which caused microorganisms

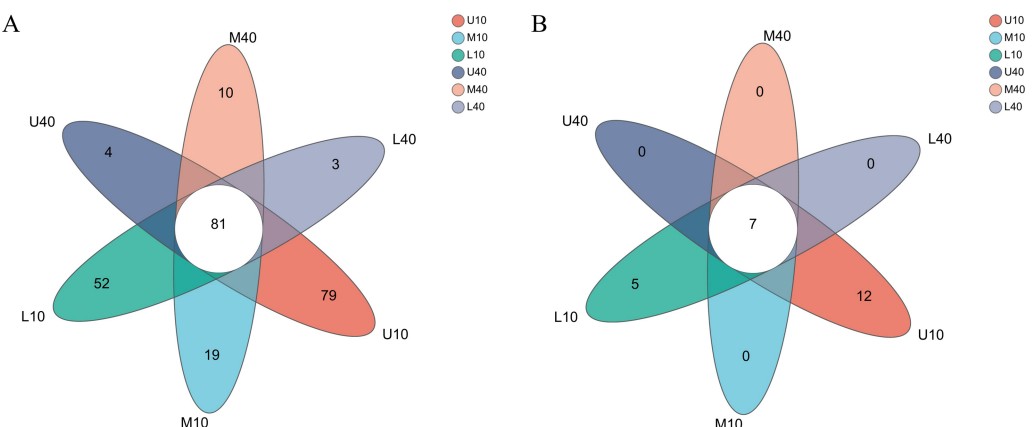

**Figure 2** Venn diagram of microbial genus levels of bacteria (A), archaea (B).

inside the PM to compete with each other to eliminate other microbes that are not beneficial for liquor brewing and retain dominant microorganisms that promote the stability of the microecological environment, thereby of the diversity of the 40-year PM decreased with fewer specific genera.

Small differences were observed in the levels of archaeal genera among the different PM samples (Fig. 2B). Seven common microbial genera were noted in all samples, and only U10 and L10 had 12 and five specific microbial genera, respectively. By contrast, the 40-year PM had no specific microbial genera. The differences in the archaeal genera within the PM were not significant. However, the lack of domestication time and the ability of some archaea to interact with bacteria not beneficial for liquor brewing were retained for the 10-year PM, so they had specific genera. With PM aging, the archaea in the interior of the 40-year PM were gradually domesticated, and therefore, they had no specific genera. The certain quality differences in the PM samples were possibly caused by the differences in the abundance of dominant archaeal and bacterial genera and specific bacterial genera, with bacteria accounting for a greater weight in the overall influence. Thus, bacteria were the factor determining PM quality, with archaea playing an auxiliary role. They together enhance the quality of Nongxiangxing baijiu (*Mu et al., 2022*), which is consistent with the results of *Zhao et al. (2022)*.

## Beta diversity of the PM

To outline the distribution of microbial communities in all samples, we conducted PCA on all samples (Fig. 3). The bacterial communities of the 10- and 40-year PM samples were clearly separated, with those of the 10-year PM distributed on the negative axis and those of the 40-year PM distributed on the positive axis (Fig. 3A). This indicated that significant differences were present between PM samples of different ages. Meanwhile, U10 was located in the two quadrant (upper left), M10 and L10 were located in the third quadrant (lower left), and the 40-year PM samples were all located in the fourth quadrant (lower right). This indicated that the spatial distribution characteristics of bacterial communities of the

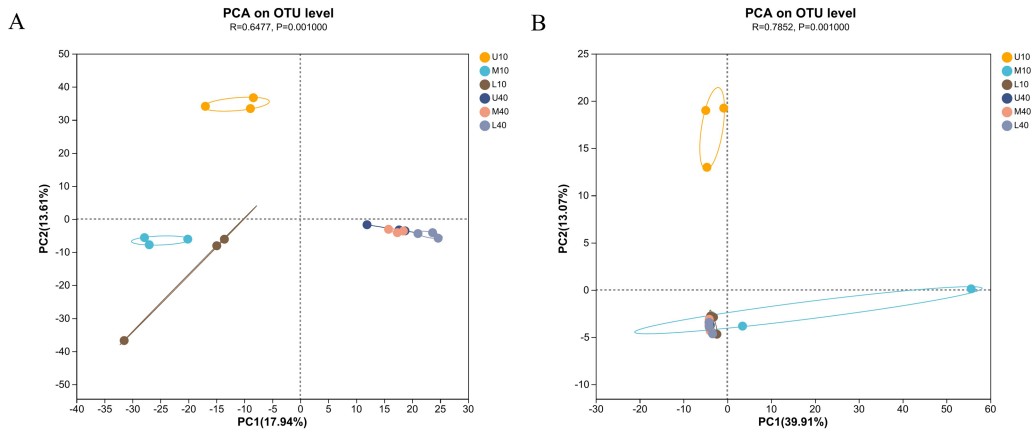

**Figure 3** PCA of microbial communities of bacteria (A), archaea (B).

40-year PM samples were more similar after a long domestication period, whereas those of the 10-year PM were still in the initial stage, with significant differences observed between the upper, middle, and lower layers. In addition, M10, L10, U40, M40 and L40 were located on the negative axis on the second principal component. This indicated that the bacterial communities of these samples were similar on the second principal component, and M10 and L10 PM samples were already in the process of transformation to mature PM. U10 was located in the two quadrant (upper left), M10 was located in the fourth quadrant (lower right), and the rest of the samples were located in the third quadrant (lower left) (Fig. 3B). U10 was located on the positive axis on the second principal component, while the remaining samples were located on the negative axis, which indicated that the archaeal communities of the samples other than the U10 sample were more similar on the second principal component. The effect of cellar age on archaea was lower than that on bacteria. We speculate that the domestication rate of archaea was higher than that of bacteria (*Fu et al., 2021*).

A total of 29 phyla, 276 families, and 540 genera of bacterial communities were identified in all of the 18 samples tested. As shown in Fig. 4A, some differences were noted in the phylum and relative abundance of different PM samples, in which the dominant phylum (relative abundance >1%) was Firmicutes (88.69%), followed by Bacteroidota (5.64%), Proteobacteria (1.98%), Actinobacteriota (1.49%), Synergistota (0.90%), Cloacimonadota (0.51%), and Verrucomicrobiota (0.20%). Among them, Firmicutes was the absolute dominant microbiome in all PM samples, with a relative abundance of 78.33%–96.40%, and the average relative abundance was the highest and exceeded 85%. The relative abundance of Firmicutes was slightly higher in the 10-year PM than in the 40-year PM, and the spatial distribution exhibited an increasing trend, followed by a decreasing trend. The relative abundance of Firmicutes was the highest in the middle layer. The relative abundance of Bacteroidota was higher in the 40-year PM than in the 10-year PM, and the spatial distribution of the 10-year PM increased with an increase in cellar depth, while the spatial distribution of the 40-year PM exhibited a trend of decrease and then increase,

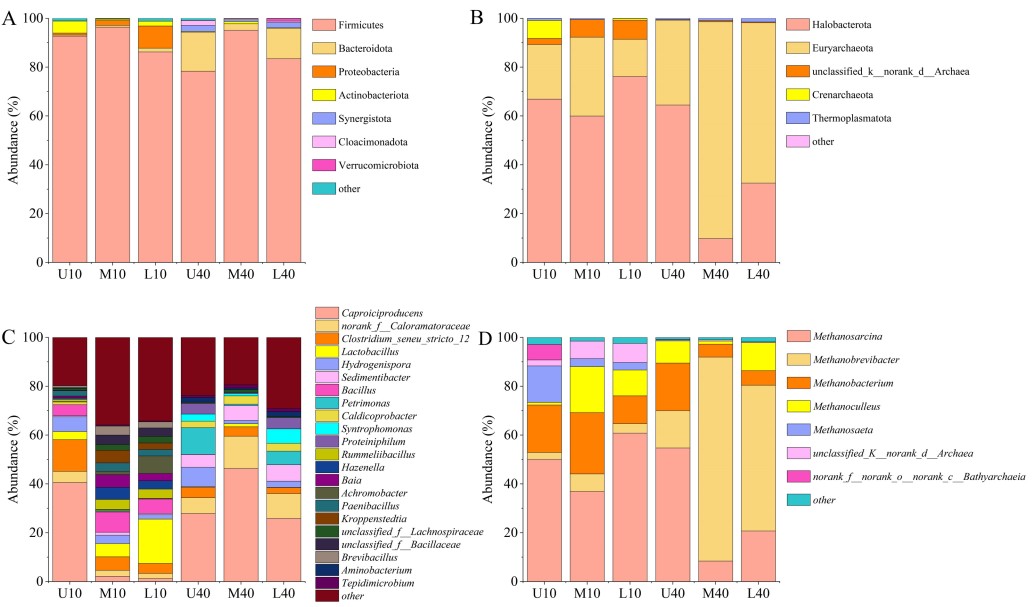

**Figure 4** Community composition of the phylum level bacteria (A), archaea (B). Community composition of the genus level bacteria (C), archaea (D).

which was opposite to the changes observed in the relative abundance of Firmicutes. Proteobacteria was mainly present in the 10-year PM and its relative abundance increased as cellar depth increased, which is consistent with the results of *Tao et al. (2014a)*. The relative abundance of Actinobacteriota was higher in the 10-year PM than in the 40-year PM, with the highest relative abundance noted in U10, indicating that Actinobacteriota was likely one of the representative bacterial phyla of the new PM.

A total of eight phyla, 24 families, and 34 genera of archaeal communities were identified in all the 18 samples tested. As shown in Fig. 4B, the phylum and abundance present in different PM samples exhibited some differences, in which the dominant phylum (relative abundance >1%) was Halobacterota (51.64%), Euryarchaeota (43.19%), unclassified_k__norank_d__Archaea (3.09%), Crenarchaeota (1.34%), followed by Thermoplasmatota (0.73%). Halobacterota and Euryarchaeota were the absolute dominant microbiomes in all PM samples, with their relative abundances ranging from 9.76% to 76.17% and 15.25% to 88.81%, respectively. The relative abundance of Halobacterota was higher in the 10-year PM than in the 40-year PM, and the spatial distribution exhibited a decreasing trend, followed by an increasing trend. The relative abundance of Euryarchaeota was higher in the 40-year PM than in the 10-year PM, and the spatial distribution displayed an increasing trend, followed by a decreasing trend. The unclassified_k__norank_d__Archaea and Crenarchaeota were mainly present in the 10-year PM, which suggests that they are the representative archaeal phyla of the new PM.

Twenty dominant genera (relative abundance >1%) were observed in the different PM samples, including *Caproiciproducens* (24.01%), *norank_f__Caloramatoraceae* (5.61%), *Clostridium_seneu_stricto _ 12* (3.39%), *Lactobacillus* (4.18%), *Hydrogenispora* (3.87%),

*Sedimentibacter* (3.38%), *Bacillus* (3.12%), *Petrimonas* (2.88%), *Caldicoprobacter* (1.81%), *Syntrophomonas* (1.73%), *Proteiniphilum* (1.59%), and *Rummeliibacillus* (1.50%) (Fig. 4C). The relative abundance of *Caproiciproducens* was the highest among all samples, accounting for more than 20%, ranging from 1.21% to 46.39%. The relative abundance of *Caproiciproducens* was higher in the 40-year PM than in the 10-year PM, and the spatial distribution of the 10-year PM decreased with an increase in cellar depth, whereas that of the 40-year PM first increased and then decreased. The relative abundance of *Clostridium_seneu_stricto_12* was higher in the 10-year PM than in the 40-year PM, and the spatial distribution exhibited a gradual decreasing trend. The relative abundance of *Lactobacillus* was significantly higher in the 10-year PM than in the 40-year PM and increased with an increase in cellar depth in the 10-year PM. The relative abundance of *Hydrogenispora* was less different between the 10- and 40-year PM. Both year samples exhibited the highest relative abundance in the upper layer, with *Hydrogenispora* capable of producing acetic acid, ethanol, and hydrogen by using glucose as a substrate. The relative abundance of *Sedimentibacter* was significantly higher in the 40-year PM than in the 10-year PM and increased with an increase in cellar depth in the 40-year PM. *Sedimentibacter* thus became a key genus in the old PM after domestication. *Bacillus* and *Rummeliibacillus* were mainly present in the 10-year PM, which indicates that they were the representative genera of the new PM. *Petrimonas* and *Proteiniphilum* were mainly present in the 40-year PM and exhibited similar patterns of variation. This suggested that they had a mutually reinforcing effect on each other and were key genera in the old PM. The relative abundance of *Caldicoprobacter* was higher in the 40-year PM than in the 10-year PM. Meanwhile, M40 and L40 were higher than U40, indicating that their relative abundances gradually increased after domestication, while U10 was higher than M10 and L10, probably because the new PM was frequently maintained. *Syntrophomonas* was mainly present in the 40-year PM and had the highest relative abundance in L40, while it was very rare in the 10-year PM.

As shown in Fig. 4D, seven dominant genera (relative abundance >1%) were present in different PM samples, including *Methanosarcina* (38.62%), *Methanobrevibacter* (28.72%), *Methanobacterium* (14.45%), *Methanoculleus* (8.71%), and *Methanosaeta* (3.63%). The relative abundance of *Methanosarcina* was the highest among all the samples, ranging from 8.43% to 60.81%. The relative abundance of *Methanosarcina* was higher in the 10-year PM than in the 40-year PM, with the spatial distribution exhibiting a decreasing trend, followed by an increasing trend. The relative abundance of *Methanobrevibacter* was significantly higher in the 40-year PM than in the 10-year PM, with the spatial distribution exhibiting an increasing trend, followed by a decreasing trend. The relative abundances of *Methanobacterium* and *Methanoculleus* were higher in the 10-year PM than in the 40-year PM, with their spatial distributions exhibiting an increasing trend, followed by a decreasing trend. *Methanosaeta* was mainly present in the 10-year PM, and its abundance decreased as cellar depth increased, indicating that *Methanosaeta* was a representative archaeal genus of the new PM.

## Differential microbes of the PM

The results of the aforementioned analysis revealed significant differences in the microbial community structure of the PM samples from different time spaces. We therefore further analyzed the differences between the microbial community structures of different samples.

Figure 5A presents the results of differential bacteria. A total of 18 species of differential microbes were detected at the genera level for all samples under the condition of LDA >4. Among these species, one species was from U10, *Clostridium_seneu_stricto_12*; six species were from M10, namely *Bacillus*, *Baia*, *Kroppenstedtia*, *Rummeliibacillus*, *Brevibacillus*, and *Paenibacillus*; three species were from L10, namely *Lactobacillus*, *Achromobacter*, and *unclassified_f__Lachnospiraceae*; two species were from U40, namely *Petrimonas* and *hydrogenispora*; three species were from M40, namely *Caproiciproducens*, *norank_f__Caloramatoraceae*, and *Caldicoprobacter*; and three species were from L40, namely *Sedimentibacter*, *Syntrophomonas*, and *Proteiniphilum*.

Figure 5B presents the results of differential archaea. A total of five species of differential microbes were detected at the genera level for all samples under the condition of LDA >2. Among these species, one species was from U10, *Methanosaeta*; two species were from L10, namely *Methanosarcina* and *unclassified_k__norank_d__Archaea*; one species was from M40, *Methanobrevibacter*; one species was from L40, *unclassified_p__Halobacterota*; and M10 and U40 had no differential microbes.

## Correlation analysis of physicochemical properties and the microbial community structure

The growth and reproduction metabolism of microbes depends to some extent on the suitability of the physicochemical properties. The differences in the physicochemical properties of PM from different time spaces cause some differences in the growth of their microorganisms (*Wu et al., 2022a*). The correlation between the physicochemical properties and dominant microbial genera of the PM from different time spaces was determined through the redundancy analysis (RDA). The total explanation of the two principal components for bacteria was 59.97% and that for archaea was 74.75% (Fig. 6). All physicochemical properties were positively correlated with *Caproiciproducens*, *norank_f__Caloramatoraceae*, *Sedimentibacter*, *Petrimonas*, *Caldicoprobacter*, *Syntrophomonas,* and *Proteiniphilum*, whereas negatively correlated with *Clostridium_seneu_stricto_12*, *Lactobacillus*, and other dominant microbial genera (Fig. 6A), which is in general agreement with the results of *Xiao et al. (2023)*. All physicochemical properties were positively correlated with *Methanobrevibacter* (Fig. 6B), indicating that this genus plays the same role in archaea as that played by *Caproiciproducens* in bacteria. Moisture, ammoniacal nitrogen, available phosphorus, and humus were negatively correlated with other dominant microbial genera. pH was positively correlated with *Methanobacterium*, *Methanosaeta*, and *norank_f__norank_o__norank_c__Bathyarchaeia*, whereas negatively correlated with *Methanosarcina*, *Methanoculleus*, and *unclassified_K__norank_d__Archaea*. Moreover, physicochemical properties were positively correlated with M40 and L40, which indicated that these properties can be used to

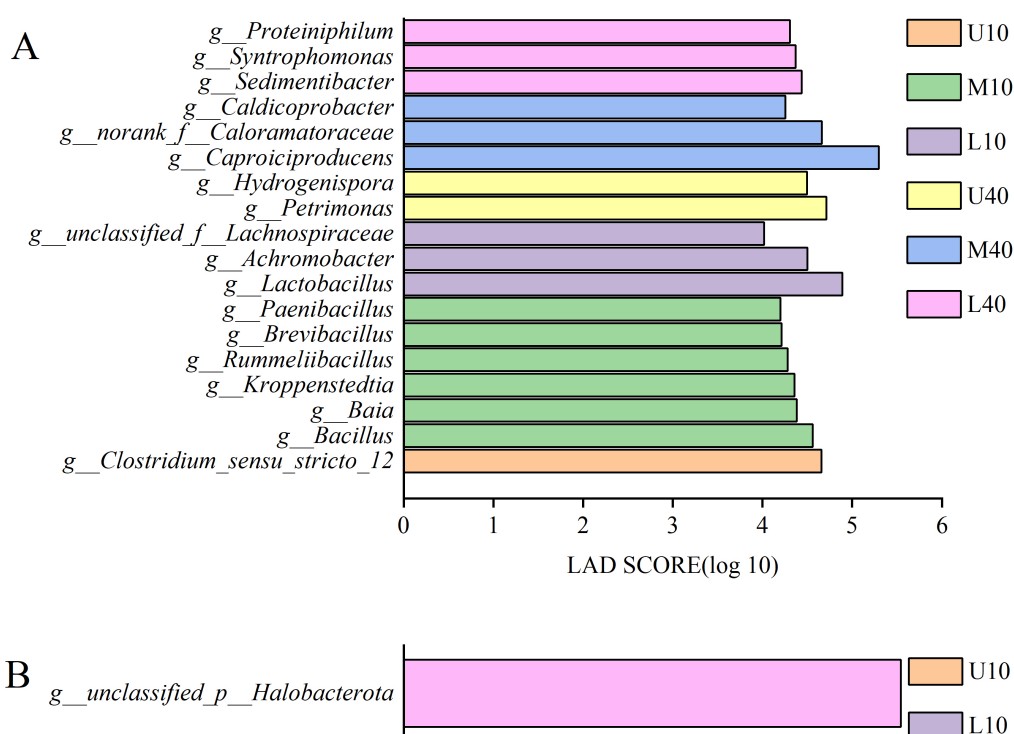

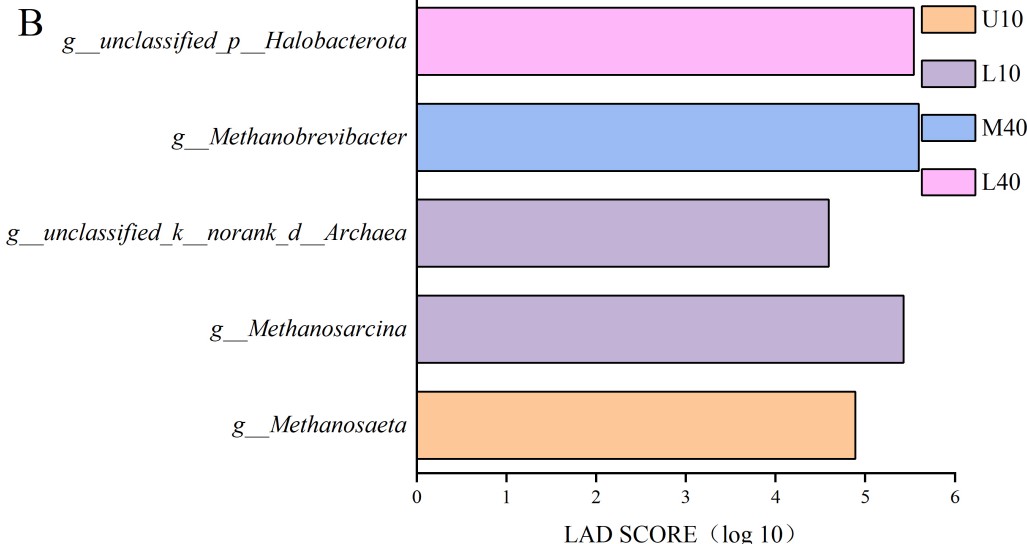

**Figure 5** LDA discriminant bar chart of bacteria (A), archaea (B).

distinguish between samples of different cellar ages and between upper and lower middle layers.

## Correlation analysis of volatile compounds and the microbial community structure

The main flavor ethyl caproate and its precursor substance hexanoic acid are very crucial for Nongxiangxing baijiu. If these components are missing or present in disproportionate amounts, this will cause an imbalance, deterioration of liquor quality, and loss of unique characteristics. Thus, hexanoic acid and ethyl caproate can be used to some extent as

Peer J

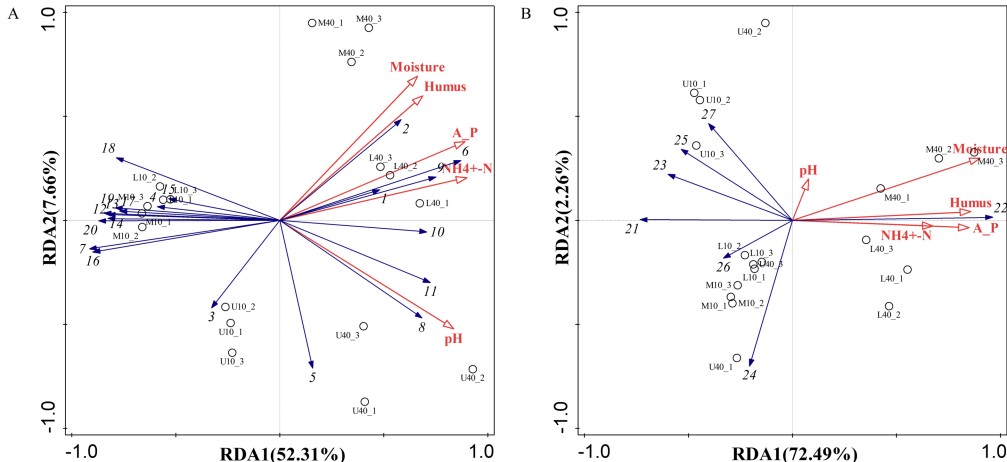

1 Caproiciproducens, 2 norank_f__Caloramatoraceae, 3 Clostridium_seneu_stricto_12, 4 Lactobacillus, 5 Hydrogenispora, 6 Sedimentibacter, 7 Bacillus, 8 Petrimonas, 9 Caldicoprobacter, 10 Syntrophomonas, 11 Proteiniphilum, 12 Rummeliibacillus, 13 Hazenella, 14 Baia, 15 Achromobacter, 16 Paenibacillus, 17 Kroppenstedita, 18 unclassified_f__Lachnospiraceae, 19 unclassified_f__Bacillaceae, 20 Brevibacillus 21 Methanosarcina, 22 Methanobrevibacter, 23 Methanobacterium, 24 Methanoculleus, 25 Methanosaeta, 26 unclassified_K__norank_d__Archaea, 27 norank_f__norank_o__norank_c__Bathyarchaeia

**Figure 6** **Redundancy of physicochemical properties and microbial genus levels of bacteria (A) and archaea (B).**

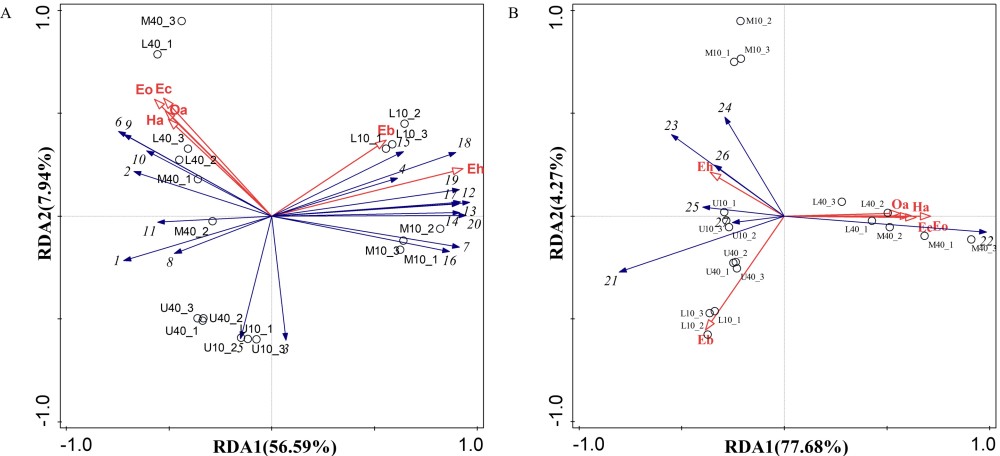

1 Caproiciproducens, 2 norank_f__Caloramatoraceae, 3 Clostridium_seneu_stricto_12, 4 Lactobacillus, 5 Hydrogenispora, 6 Sedimentibacter, 7 Bacillus, 8 Petrimonas, 9 Caldicoprobacter, 10 Syntrophomonas, 11 Proteiniphilum, 12 Rummeliibacillus, 13 Hazenella, 14 Baia, 15 Achromobacter, 16 Paenibacillus, 17 Kroppenstedita, 18 unclassified_f__Lachnospiraceae, 19 unclassified_f__Bacillaceae, 20 Brevibacillus 21 Methanosarcina, 22 Methanobrevibacter, 23 Methanobacterium, 24 Methanoculleus, 25 Methanosaeta, 26 unclassified_K__norank_d__Archaea, 27 norank_f__norank_o__norank_c__Bathyarchaeia

**Figure 7** **Redundancy of volatile compounds and microbial genus levels of bacteria (A) and archaea (B), note: Ec: ethyl caproate, Eo: ethyl octanoate, Ha: hexanoic acid, Oa: octanoic acid, Eb: ethyl butyrate, Eh: ethyl hexadecanoate.**

indicators for assessing the quality of PM and Nongxiangxing baijiu. Differential volatile compounds were selected from PM samples from different time spaces to perform RDA with dominant microbial genera. The RDA results are presented in Fig. 7. The total

explanation of the two principal components for bacteria was 64.53% and that for archaea was 81.95%.

*Caproiciproducens*, *norank_f__Caloramatoraceae*, *Sedimentibacter*, *Petrimonas*, *Caldicoprobacter*, *Syntrophomonas*, and *Proteiniphilum* were positively correlated with ethyl caproate, ethyl octanoate, hexanoic acid, and octanoic acid, whereas negatively correlated with ethyl butyrate and ethyl hexadecanoate (Fig. 7A). The other dominant microbial genera were basically positively correlated with ethyl butyrate and ethyl hexadecanoate, whereas negatively correlated with ethyl caproate, ethyl octanoate, hexanoic acid, and octanoic acid. *Methanobrevibacter* was positively correlated with ethyl caproate, ethyl octanoate, hexanoic acid, and octanoic acid, whereas negatively correlated with ethyl butyrate and ethyl hexadecanoate (Fig. 7B). The other dominant microbial genera were positively correlated with ethyl butyrate and ethyl hexadecanoate, whereas negatively correlated with ethyl caproate, ethyl octanoate, hexanoic acid, and octanoic acid. Ethyl caproate, ethyl octanoate, hexanoic acid, and octanoic acid exhibit the same physicochemical properties and display positive correlations with M40 and L40.

## DISCUSSION

In this study, the physicochemical properties, volatile compounds, and microbial community structure of PM from different time spaces were comparatively analyzed. The relationship between these three indicators was investigated through LEfSe analysis and RDA. The results unveiled significant differences between the PM samples from different time spaces.

The physicochemical properties of the 40-year PM were significantly higher than those of the 10-year PM. The pH in the upper PM was higher than that in the middle and lower PM. This was possible because the grains fermented in the cellar, thereby producing free water through microbial decomposition and metabolism, leaching out the nutrients from the grains, and forming a brownish-yellow, slightly viscous acidic turbid liquid yellow water, which gradually penetrated into the bottom of the cellar. Meanwhile, the pH of the 40-year PM was closer to the growth conditions of *Caproiciproducens*, which is conducive to hexanoic acid production (*Wang et al., 2021*). Being the main nitrogen source in PM, ammoniacal nitrogen can determine whether microbial reproduction is vigorous. Studies have reported that ammoniacal nitrogen was higher in high-quality cellars than in ordinary cellars, and the present study results are consistent with those of past studies (*Hu et al., 2016*). The humus content was significantly different in all samples, which indicated that humus significantly affects PM quality. As PM matures and microorganisms domesticate and increase in abundance, organic matter is gradually decomposed into humus, thereby increasing the humus content (*Wu et al., 2018*). When the cellar age and depth increase, changes occur in the brewing environment, such as anaerobic degree; microorganisms perform metabolic activities according to their adaptations; and numerous disadvantageous microorganisms not conducive to liquor brewing die and are deposited, which increases the content of elemental phosphorus (*Zhang et al., 2020d*). Moreover, the study results revealed that the available phosphorus in all 10-year PM samples reached the standard of

secondary PM and all the 40-year PM samples reached the standard of primary PM (*Zhao et al., 2012*).

Ethyl caproate is among the main volatile compounds that constitute the unique characteristics of Nongxiangxing baijiu, with a fruity flavor. The content of ethyl caproate is crucial for the quality of this liquor. According to related studies, ethyl octanoate is second only to ethyl caproate in flavor intensity, and sometimes even exceeds it, with a sweet and fruity flavor. This indicates that the contribution of ethyl octanoate to the flavor of Nongxiangxing baijiu is equally important (*Xu et al., 2022b*). The contents of ethyl caproate and ethyl octanoate in the 40-year PM were higher than those in the 10-year PM, which indicated that the PM was gradually domesticated and its quality improved with an increase in the cellar use time. Ethyl butyrate is fruity and can contribute a refreshing flavor within a certain level, whereas, when present in excess, can cause the appearance of an unpleasant sour odor. In the meantime, ethyl butyrate only appears in the lower PM, suggesting that ethyl butyrate production is related to the microecological environment of the lower cellar part and the degree of anaerobiosis. Hexanoic acid and octanoic acid are precursors of ethyl caproate and ethyl octanoate, respectively, which have a sweaty flavor and can synthesize corresponding esters with alcohols through esterification to improve the flavor quality of Nongxiangxing baijiu (*Wang et al., 2022*). The contents of hexanoic acid and octanoic acid in M40 and L40 were higher than those in M10 and L10, which may also be the reason for the higher contents of ethyl caproate and ethyl octanoate. N-hexanol has a green flavor and can be converted to the corresponding hexyl ester compounds, and it was only present in the 40-year PM. This may be the reason why the hexyl caproate content was higher in the 40-year PM than in the 10-year PM. Moreover, 4-methylphenol was present only in the 40-year PM, which has animal flavor. The content of 2, 4-di-tert-butylphenol was higher in the 10-year PM than in the 40-year PM and has a phenol flavor. Phenols present in an appropriate amount can increase the aftertaste of baijiu, whereas excessive phenols can produce an unpleasant flavor.

The diversity and richness of bacteria were higher in the 10-year PM than in the 40-year PM, possibly because microbes in new cellars were still in the initial stage and microbes both useful and useless for liquor brewing existed in the PM. Microorganisms in the 40-year PM gradually transformed into functional microbes adapted to the liquor brewing environment through continuous stress and domestication after long-term use, which caused changes in the microbial community structure and resulted in a stable microecological environment for liquor brewing (*Huang et al., 2014*; *Xiao et al., 2023*). The change in the spatial level of the 10-year PM may have occurred due to the artificial cellar maintenance. The maintenance solution is generally poured from the upper layer to the lower layer, and so, microbial diversity first exhibits a trend of increase. However, with a gradual increase in cellar depth, the anaerobic degree also increases gradually. Thus, microorganisms in the maintenance solution do not grow and reproduce adequately, and a trend of decline is noted. Furthermore, the domestication rate of L10 microorganisms was higher than that of U10, which transformed the microorganisms into functional microorganisms that are more conducive to brewing, thereby resulting in a decline in richness. The opposite result was noted in the spatial distribution of the 40-year PM because

the PM reached maturity after a long period of domestication. L40 had a more dominant brewing microbiome and functional microorganisms in a strictly anaerobic environment, so, the diversity and richness of L40 were higher than those of U40, which also confirms that the baijiu quality of the lower cellar was higher than that of the upper cellar. In addition, the $\alpha$ diversity index of archaea indicates that the growth and reproduction patterns of bacteria and archaea are similar. Moreover, they exhibit mutually beneficial symbiosis to jointly maintain the microecological environment of PM, promote the production of key volatile compounds, and enhance the quality of baijiu. Archaea in the PM are mostly methanogenic bacteria that require a more demanding anaerobic environment for reproduction. This explains why the diversity and richness of L40 were higher than that of U40.

At the phylum level, Firmicutes serves as an acid and aroma producer in the PM. Some of these microorganisms can metabolize and produce hexanoic acid, which is further esterified with ethanol to synthesize ethyl caproate, the core volatile compound in Nongxiangxing baijiu. Therefore, Firmicutes is the key microbial phylum in the PM (*Bautista-Gallego et al., 2014*). The relative abundance of Firmicutes of U10 and M10 were higher than that of of L10 because of maintenance, whereas the 40-year PM was lower than the 10-year PM probably because the relative abundance of other phyla was enhanced through domestication, and the baijiu quality was generally higher in the 40-year PM than in the 10-year PM. This indicates that although Firmicutes dominates the microecological environment of the PM, it still needs to interact with other phyla to improve PM quality. Bacteroidota can reduce iron ions and provide maturity characteristics to the PM, while some of these microorganisms can metabolize and produce acetic acid and propionic acid (*Ren et al., 2023*; *Yu et al., 2021*). The methanogenic bacteria belonging to Euryarchaeota can symbiotically grow with *Caproiciproducens* and remove hydrogen inhibition through an interspecific "hydrogen transfer" effect, thereby promoting *Caproiciproducens* growth and metabolism and increasing the ethyl caproate content in Nongxiangxing baijiu (*Chen et al., 2020*). This is possibly a reason for the high ethyl caproate content in the 40-year PM, and this result also explains the speculation that Firmicutes need to interact with other phyla to improve PM quality.

At the genus level, *Caproiciproducens* is among the key microorganisms involved in producing organic acids, such as lactic acid, acetic acid, butyric acid, and hexanoic acid, through chain extension reactions, as well as in converting lactic acid into hexanoic acid, the precursor substance of ethyl caproate (*Chai et al., 2021*). This suggests that *Caproiciproducens* are among the main reasons for the higher hexanoic acid content of the 40-year PM than of the 10-year PM. However, in terms of spatial distribution, the hexanoic acid content did not correspond to the relative abundance of *Caproiciproducens*, suggesting that other genera interacted synergistically to produce hexanoic acid. The variation pattern of *norank_f__Caloramatoraceae* was consistent with that of *Caproiciproducens*, suggesting that *norank_f__Caloramatoraceae* is a representative genus of matured PM. Microbes belonging to the genus *Clostridium* usually grow, metabolize, and reproduce under anaerobic or partly anaerobic conditions, with some of them capable of converting organic matter into hexanoic acid and butyric acid, which are further esterified with ethanol to synthesize ethyl caproate and ethyl butyrate, respectively (*Hu et al., 2021b*; *Su et al., 2019*;

*Yu et al., 2021*). The relative abundance of *Clostridium_seneu_stricto_ 12* was higher in the 10-year PM than in the 40-year PM. This suggests that *Clostridium_seneu_stricto_ 12* possibly plays a better role in producing hexanoic acid and butanoic acid in the new PM than in the old PM, and that *Clostridium_seneu_stricto_ 12* was may not the main hexanoic acid and butanoic acid-producing microbial genera in the old and lower PM. The results indicated that *Lactobacillus* was among the signature genera of the new PM and was the key to its quality. Our results study are consistent with those of *Liu et al. (2022c)*, and *Lactobacillus* can thus be used as an evaluation index for new PM. When fermentation proceeds, yellow water, the by-product of brewing grains, gradually accumulates at the bottom of the cellar. This yellow water usually has a low pH and strong acidity, which is conducive to *Lactobacillus* growth and reproduction, but not to the survival of other non-acidophilic microorganisms. Moreover, some *Lactobacillus* spp. can degrade lactic acid to produce acetic acid under certain pH conditions (*Zhang et al., 2020c*). Related studies have revealed that *Petrimonas* and *Proteiniphilum* are commonly present in old PM and can metabolize sugars to synthesize acetic acid, propionic acid, and other volatile compounds of Nongxiangxing baijiu while associating symbiotically with methanogens in mature PM (*Hu et al., 2021a*; *Liu et al., 2022b*; *Ren et al., 2023*; *Wu et al., 2022b*; *Zhang & Deng, 2019*). The results of the present study are consistent with those of the aforementioned studies. *Syntrophomonas* can degrade long-chain fatty acids and produce acetic acid, propionic acid, butyric acid, and hydrogen through direct interspecies electron transfer, while it can symbiotically convert hydrogen to methane along with hydrogenotrophic methanogens. This relieves the inhibition of *Caproiciproducens* and promotes the metabolic operation of the complete PM microbiome (*Chai et al., 2020*; *Gao, Wu & Zhang, 2020*; *Meng et al., 2021*; *Vanwonterghem et al., 2014*). The relative abundance of *Syntrophomonas* was the highest in L40, which indicated that it is a key genus in old PM and is among the reasons for the old PM and lower PM being superior to new PM and upper PM, respectively. *Methanosarcina* can use both hydrogen and acetic acid, and *Methanobrevibacter*, *Methanobacterium*, and *Methanoculleus* can use only hydrogen. These archaea can interact with *Caproiciproducens* to increase the ethyl caproate content through interspecific hydrogen transfer (*Fu et al., 2021*; *Tao et al., 2014b*). Therefore, the relative abundance of the main archaeal genera in the PM at different time spaces exhibited differences, possibly because of some substrate competition and complementary interactions between the genera as they use the same substrate for increasing the ethyl caproate content through interspecific hydrogen transfer.

In addition, some genera with <1% relative abundance were also important, and both *Aminobacterium* and *Tepidimicrobium* were mainly present in the 40-year PM. The relative abundance of both was the highest in L40. According to related studies, *Sedimentibacter* and *Aminobacterium* can metabolically degrade amino acids to produce ammoniacal nitrogen as a nitrogen source for microbial growth and reproduction in PM, and can further synthesize acetic acid and butyric acid (*Fang, Wang & Yan, 2023*; *Hu et al., 2021b*; *Meng et al., 2021*). *Caldicoprobacter* and *Tepidimicrobium* can metabolize glucose, galactose, and fructose as substrates to synthesize lactic acid, acetic acid, and butyric acid, which serve as precursors for hexanoic acid synthesis (*Liu et al., 2017*; *Wu et al., 2022b*). This fully illustrates the significance of rare microbial genera and that an extremely complex microecological

environment is present inside the cellar. With the continuous domestication and aging of the PM, dominant and functional microorganisms adapt to the brewing environment and survive in the old PM. However, the quality of PM and baijiu does not depend on the absolute dominant microorganisms present in the PM. Many functional microorganisms and rare microbiomes with the absolute dominant microbiome to improve the quality of PM and baijiu through interspecific interactions (*Zhao et al., 2022*). For example, M40 had the highest relative abundance of *Caproiciproducens*, but it had a significantly lower content of major volatile compounds, such as hexanoic acid and ethyl caproate, than L40. This further suggests that our speculation is accurate and that a considerable number of microorganisms are present inside the cellar, including those brought by the environment, the Daqu, and the fermented grain system. These microbes function together to maintain the stability of the microecological environment inside the cellar. Interestingly, the relative abundance of *Caproiciproducens* and some functional microorganisms was higher in U10 than in M10 and L10, which contradicts the results of some studies. This is possibly due to the artificial maintenance of the new cellar as the maintenance solution is generally poured from the upper layer. This contradiction also was possibly related to the physicochemical properties of the PM, some unknown growth factors, vitamins, minerals, and other trace elements of the PM, and environmental conditions of the PM.

Of note, ethyl caproate, the main volatile compound of Nongxiangxing baijiu, is not simply produced through the esterification of hexanoic acid with ethanol produced by *Caproiciproducens*, but the esterification enzyme content and some other unknown factors also affect its synthesis. Related studies have shown that Daqu selectively esterifies volatile acids produced by the PM microorganisms. Daqu was mainly esterified in fermented grains to synthesize ethyl caproate, but barely converts lactic acid and butyric acid into corresponding ethyl esters. This indicates that ethyl caproate in fermented grains was mainly esterified by Daqu esterase. By contrast, ethyl lactate and ethyl butyrate were mainly esterified through non-enzymatic catalysis, and these esters migrate from the fermented grains to the PM (*Gao et al., 2021*). Moreover, microbial species also migrate between the Daqu and fermented grains and the PM because of the chemotaxis action of the strains, which allows them to interact with each other (*Wu et al., 2022a*).

Our LEfSe analysis of the microbial community structure of the PM samples from different time spaces revealed that differential microorganisms led to significant differences in PM quality. The results were consistent with those of the microbial community composition analysis. *Clostridium_seneu_stricto_12* was the differential microorganism of U10, and this further explains our previous speculation that it is one of the hexanoic acid and butyric acid-producing microbial genera in the new PM. Some *Bacillus* spp. can produce esters, but they were mainly distributed in the 10-year PM. This suggests that microbes capable of esterifying and synthesizing the main flavors of Nongxiangxing baijiu and those producing off-odors through esterification are both present in new PM, and microbes producing off-odors may account for a larger proportion. *Lactobacillus* was the differential microorganism of L10. *Caproiciproducens* and *Lactobacillus* are known to have a mutually inhibitory effect on each other, thereby weakening the growth and reproduction of the other microbial genus when the relative abundance of one genus exceeds a certain

range (*Andersen et al., 2017*; *Li et al., 2022*; *Zhang et al., 2020d*). This suggests that hexanoic acid is somewhat correlated to lactic acid, which may be because *Caproiciproducens* uses lactic acid from *Lactobacillus* to produce hexanoic acid (*Liu et al., 2022a*; *Wang et al., 2021*). Moreover, *Zhang et al. (2020b)* showed that *Caproiciproducens* in the surface layer of old PM can act as a protective layer, thereby preventing the entry of *Lactobacillus* and promoting anaerobic bacteria growth. Meanwhile, old PM forms a complete metabolic chain to promote the degradation of lactic acid and other acids, whereas new PM does not form this metabolic chain. Therefore, the relationship between the utilization and constraints of these two genera is speculated to be a reason why the hexanoic acid content in old PM is higher than that in new PM. The differential microorganisms in the 40-year PM were basically key functional microbial genera. Except for *Caproiciproducens*, other microbial genera were less relative abundance, but they all played the role of producing acetic acid, propionic acid, butyric acid, and other small-molecule organic acids to offer precursors for *Caproiciproducens*. Consequently, the acid content of M40 and L40 was significantly higher than that of M10 and L10. *Caproiciproducens* was the differential microorganism of M40. The volatile compounds indicated that the hexanoic acid content was significantly higher in L40 than in M40, which further confirmed our speculation that *Caproiciproducens* has a dominant role in the PM and exhibits a mutually beneficial symbiosis with other low-abundance microbiomes. The synergistic effect produces small-molecule organic acids such as hexanoic acid. *Sedimentibacter*, *Syntrophomonas*, and *Proteiniphilum* were probably the main genera of the so-called low-abundance microbiome in L40. Bacteria involved in hexanoic acid and butyric acid synthesis in Nongxiangxing baijiu in Sichuan and other production areas were different, with *Ruminiclostridium* and *Syntrophomonas* observed in Sichuan and *Caproiciproduces* and *Clostridium* observed in Henan (*Gou et al., 2020*; *Liu et al., 2022c*). *Methanobrevibacter* was a differential microorganism of M40, while the highest relative abundance of *Caproiciproducens* in M40 suggests that *Methanobrevibacter* has a mutually beneficial symbiosis with *Caproiciproducens*, thereby promoting mutual reproduction. The *unclassified_p__ Halobacterota* was a differential microorganism of L40. According to previous studies, methanogenic bacteria that use hydrogen or hydrogen and acetic acid as substrates belong to the archaebacterial genera that mainly functions in PM, whereas *unclassified_p__Halobacterota* was embodied in L40. This indicates that *unclassified_p__Halobacterota* is possibly one of the signature archaebacterial genera in old PM. It catalyzes the aging of PM, relieves the hydrogen inhibition of *Caproiciproducens*, promotes the production of hexanoic acid and other organic acids, thus enhancing the content of related esters, and maintains the stability of the microecological environment jointly with other microbiome. However, these effects are only based on our speculations and need to be verified through specific studies.

The 40-year PM was domesticated and had evolved over a long period to form the dominant differential microorganisms, which is its most obvious difference from the 10-year PM. The key differential microorganisms in the 40-year PM can be used as one of the evaluation indicators for new and old PM. The new PM needs to be domesticated over a long period to form these microorganisms, thereby improving its quality. However, studies have demonstrated that the microecological environment inside the cellar is considerably more

complex than expected, and therefore, the aforementioned low-abundance microbiome, rare microbiome, unknown growth factors, and trace elements inside the cellar all play a crucial role in PM aging and the whole liquor brewing process (*Lu et al., 2021*).

The RDA results revealed that the increase in the physicochemical properties favored the reproduction and metabolism of *Caproiciproducens*, *norank_f__Caloramatoraceae*, *Sedimentibacter*, *Petrimonas*, *Caldicoprobacter*, *Syntrophomonas*, *Proteiniphilum*, and *Methanobrevibacter*, while these genera were positively correlated with ethyl caproate, ethyl octanoate, hexanoic acid, and octanoic acid. This suggests that these genera are the main functional genera in the PM and form synergistically symbiotic to play a role in hexanoic acid and octanoic acid production. In addition, the high relative abundance of *Lactobacillus* resulted in a large accumulation of lactic acid, which lowered the pH and was negatively correlated with pH. Overall, *Caproiciproducens*, *norank_f__Caloramatoraceae*, and *Methanobrevibacter* play an indispensable role in the PM. We speculate that these three genera were in a dominant position for improving the physicochemical properties and volatile compounds of the PM.

By measuring physicochemical properties, volatile compounds, and microbial community structure (the three indicators) and conducting LEfSe analysis and RDA, we found that the quality of the 40-year PM was higher than that of the 10-year PM and that the quality of lower PM was higher than that of the upper PM. This is consistent with the results of previous studies. These three indicators undergo obvious changes as the PM is gradually domesticated. PM quality is determined together by these three indicators, and a mutual influence and weighting relationship also exists among these indicators.

## CONCLUSIONS

In summary, this study systematically determined the physicochemical properties, volatile compounds, and microbial community structure of PM from different time spaces. By conducting different analyses, we concluded that the quality of the 40-year PM was higher than that of the 10-year PM, and the lower PM was higher than the upper PM. This indicated that the PM was domesticated and gradually shifted to higher maturity during the process of use, and the quality gradually improved accordingly. A comprehensive explanation of the dominant and rare microorganisms in the PM was shared to offer some theoretical basis for revealing the complex microecological environment in the cellar. We also elaborated on the specific roles of different microorganisms among PM having different qualities and clarified that some microbial genera can, to some extent, serve as indicators of PM aging, which provides some theoretical references for shortening the aging time of new PM, maintenance of old PM, and cultivation of artificial PM.

### Funding
This study was supported by the study on key technology of quality improvement of Nongxiangxing baijiu in western Sichuan (NO. HX2021034), Horizontal Cooperation

Project. The funders had no role in study design, data collection and analysis, decision to publish, or preparation of the manuscript.

## Grant Disclosures

The following grant information was disclosed by the authors:
Nongxiangxing baijiu in western Sichuan: HX2021034.
Horizontal Cooperation Project.

## Competing Interests

The authors declare there are no competing interests.

Jiaxu Zhang and Jing Luo are employed by Chengdu Shuzhiyuan Liquor Industry Co., Ltd.

## Author Contributions

- Baolin Han conceived and designed the experiments, performed the experiments, analyzed the data, prepared figures and/or tables, authored or reviewed drafts of the article, and approved the final draft.
- Hucheng Gong conceived and designed the experiments, performed the experiments, analyzed the data, prepared figures and/or tables, authored or reviewed drafts of the article, and approved the final draft.
- Xiaohu Ren analyzed the data, authored or reviewed drafts of the article, and approved the final draft.
- Shulin Tian analyzed the data, authored or reviewed drafts of the article, and approved the final draft.
- Yu Wang analyzed the data, authored or reviewed drafts of the article, and approved the final draft.
- Shufan Zhang analyzed the data, authored or reviewed drafts of the article, and approved the final draft.
- Jiaxu Zhang analyzed the data, authored or reviewed drafts of the article, and approved the final draft.
- Jing Luo analyzed the data, authored or reviewed drafts of the article, and approved the final draft.

## Data Availability

The HiSeq sequencing data are available at the Sequence Read Archive (SRA) of NCBI: PRJNA1010531.

## Supplemental Information

Supplemental information for this article can be found online at http://dx.doi.org/10.7717/peerj.17000#supplemental-information.

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
