# Peer review of "Analysis of the differences in physicochemical properties, volatile compounds, and microbial community structure of pit mud in different time spaces"

_PeerJ, doi:10.7717/peerj.17000_

## Round 0.1 · original submission · Major Revisions

- Please provide a detailed methodology.
- Connect the explanations with the provided figures; there are some missing explanations.
- Conduct a discussion based on the evidence and findings in the present study.
- Please review the English and complete the statement.

**Language Note:** The Academic Editor has identified that the English language must be improved. PeerJ can provide language editing services - please contact us at copyediting@peerj.com for pricing (be sure to provide your manuscript number and title). Alternatively, you should make your own arrangements to improve the language quality and provide details in your response letter. – PeerJ Staff

Reviewer 1 ·

Basic reporting

1. Line 46: Please unify the indented format of the text.

2. Line 111: “N” is not a universal abbreviation of year. There should be a space between the number and units, please check the similar problem in the whole manuscript.

3. Line 111-113: Are all the sampling points on each layer of PM the central point of the pit plane? How to ensure the representativeness of the sampling points in each layer?

4. Line 168-169: The sentence structure is incomplete.

5. Line 180-183: Please revise this sentence to avoid ambiguity.

6. Line 200: This statement is inconsistent with Figure 3, please check it.

7. Line 207-208: Please rewrite this statement whose subject is incorrect and check the same mistake in this manuscript

8. Line 213: “The content of L40 hexanoic and octanoic acids……”, this is not in line with the English expression.

9. Line 233: Change “Fig. 2(A, B)” to “Fig. 5 and 6”. The cited numbers of figures are inconsistent with the actual figure number.

10. Line 251-255: In this statement, the analysis and conclusion do not seem to correspond.

11. Line 280: “richness increased” compared to the 10 years PM? It seems incorrect according to the results of Ace and Chao1 indices in Table 2.

12. Line 287-290: How can the authors conclude that? Please give a more detailed analysis.

13. Line 299: The upper left space belongs to the second quadrant.

14. Line 313-314: Please add a relative reference.

15. Line 318-320: The relative abundance of each phylum was corresponding to which sample? OR Mean value?

16. Line 351: The relative abundance of Caproiciproducens was not highest in M10 and L10. The same mistake happens in the relative abundance of Methanosarcina (Line 375).

17. Line 456: Try not to use the first person.

18. Line 501: What was compared? Firmicutes? “other reasons”, this expression is over vague.

19. Line 553: How can these interspecific interactions be manifested? By competing for hydrogen?

20. Line 586: How do the authors define this “better”?

21. Line 676-677: According to the abundance and diversity analysis, the microbial community structure does not show that 40 years PM was significantly better than 10 years PM, and the lower PM was significantly better than the upper PM.

22. Line 827: When referring to a book, the detailed pages of the cited content should be added.

Experimental design

Please see the above commments.

Validity of the findings

Please see the above comments.

Additional comments

Please see the above comments.

Reviewer 2 ·

Basic reporting

The volatile compounds and microbial community structure of 10-year-old and 40-year-old pit mud (PM) and their different spatial positions were determined by headspace solid phase microextraction, gas chromatography and high throughput sequencing, and their basic physicochemical properties were also determined. The effects of different time and space on PM quality were systematically revealed, which provided theoretical basis for improving degraded PM, promoting PM aging, cellar maintenance and artificial PM cultivation in the later stage. The article is innovative, the technical route is feasible, and it has certain scientific research value. However, there are many format errors in this paper, which need to be completely revised. In addition, other issues are as follows:
1 The preface of the article is less about the previous research progress. It is suggested to increase and then refine the scientific problems to be solved in the article.
2 In the preface of the article, it is mentioned that pit mud is rich in bacteria, archaea and fungi, and in the following lines 70-77, it is indicated that the microbial community of pit mud is selected as the research index, but only bacteria and archaea are studied in the research method, and the change law of fungi in pit mud is not studied. Please supplement the explanation.
3 The expression of pit mud sampling in different spaces is vague, and it can not be seen that the samples are representative. Please explain differently.
4 In many places in the article, the graph does not match the description of the article. Please combine the result graph or modify the description of the result graph in the article.
5 In line 180-183 of the article, the description of the results of physical and chemical properties is mentioned here. The physical and chemical properties increase with the change of time and space. Is the increase of physical and chemical indicators the higher the better ? What are the evaluation criteria ? Please modify.
6 Results 189-194 lines, from Figure 1 ( A, B ) can not see that there are 40 compounds, please add and modify.
7 In the result part, the specific content of a specific compound cannot be seen in the heat map 1C, please explain and modify.
8 In the result section, line 249-260 explains the microbial diversity of pit mud at different times. The explanation of the results should belong to the discussion section, please modify it.
9 Combine Table 2 and annotate the characters in the table.
10 In Figure 12-13, the level name of microorganism should be italicized, please modify it.

Experimental design

no comment

Validity of the findings

no comment

---

## Round 0.2 · Minor Revisions

In materials and methods, please indicate the company names for specific chemicals used in the study.

For concentrations, the percentage should be indicate that it is %(w/v) or (w/w).

Please check the reference format cited in text (authors (year)).

Reviewer 2 ·

Basic reporting

Revise (Fig.2 (B)) to (Fig.2B), similar revisions need to be changed within the scope of the manuscript.

Experimental design

no comment

Validity of the findings

no comment

Additional comments

no comment

---

## Round 0.3 · accepted · Accept

The authors improved the manuscript and is now ready for publication.